# Synthesis and Characterization of Sn/Ni Single Doped and Co–Doped Anatase/Rutile Mixed–Crystal Nanomaterials and Their Photocatalytic Performance under UV–Visible Light

Qin Qin [1], Juan Wang [2], Yangwen Xia [2], Daixiong Yang [2], Qin Zhou [2], Xiaodong Zhu [2,*] and Wei Feng [2,*]

[1] Intelligent Manufacturing College, Chengdu Jincheng College, Chengdu 611731, China; stickerandqq@163.com

[2] College of Mechanical Engineering, Chengdu University, Chengdu 610106, China; wangjuan9760@163.com (J.W.); x1278704108@163.com (Y.X.); yangdaixiong1998@163.com (D.Y.); cdujxgcxysl@163.com (Q.Z.)

* Correspondence: xiaodangjia21@126.com (X.Z.); fengwei233@126.com (W.F.)

**Abstract:** Pure and Sn/Ni co–doped $TiO_2$ nanomaterials with anatase/rutile mixed crystal were prepared and characterized. The results show that pure $TiO_2$ is a mixed crystal structure composed of a large amount of anatase and a small amount of rutile. Sn doping promotes the phase transformation from anatase to rutile, while Ni doping inhibits the transformation. Both single doping and co–doping are beneficial to the inhibition of photoinduced charge recombination. Sn doping shows the best inhibitory effect on photogenerated charge recombination, and increases the utilization of visible light, displaying the highest photocatalytic activity. The decolorization degree of methylene blue (MB) by Sn–$TiO_2$ is 79.5% after 150 min. The reaction rate constant of Sn–$TiO_2$ is 0.01022 $min^{-1}$, which is 5.6 times higher than pure $TiO_2$ (0.00181 $min^{-1}$).

**Keywords:** photocatalytic activity; $TiO_2$; mixed crystal; Sn/Ni co–doping; sol–gel method

## 1. Introduction

Employing photocatalytic technology to degrade harmful substances is a feasible way to solve the problem of environmental pollution. Among the numerous photocatalysts, $TiO_2$ has attracted the most attention and has been widely studied [1–5]. The lack of visible light utilization and quantum efficiency of pure $TiO_2$ limit its photocatalytic activity [6,7]. Metal ion doping can form impurity level in the band gap and increase visible light absorption, which is one of the most commonly used methods in $TiO_2$ modification [8–13]. Krishnakumar et al.'s work shows that the grain size and energy gap are reduced and the recombination of photogenerated pairs is suppressed by Cu doping, improving the photocatalytic activity [8]. Multiple elements co–doping could yield a cooperation effect in improving the performance of $TiO_2$, obtaining better modification effect than single–element doping [14–17]. Kalantari et al. [16] found Fe and N co–doping develops a cooperation effect on improving the utilization of visible light because N doping move the valence band upward, and Fe doping introduces impurity levels below the conduction band.

It is generally believed that rutile exhibits lower photocatalytic activity than anatase due to its small specific surface area, poor adsorption performance, and few surface defects [18]. When rutile and anatase form a mixed crystal, since the mixed crystal structure will advance the transfer of photoinduced charges at the two–phase interface, it shows higher activity than single crystal [19–23].

Rutile is a thermodynamically stable phase and anatase will gradually transform into rutile with temperature rising. Therefore, anatase/rutile mixed crystal $TiO_2$ may be obtained within a certain temperature range [24,25]. The two–phase composition has an important influence on the photocatalytic property of anatase/rutile mixed crystal. It is a common method to control the relative content of anatase and rutile in the mixed crystal by heat treatment temperature and holding time [26–28]. Elsellami et al. [27] found that the mixed crystal $TiO_2$ composed of anatase 96% and rutile 4% shows the highest photocatalytic activity, which was heat treated at 600 °C. In addition, the two–phase composition can be controlled by regulating the ratio of reactants [29,30]. Li et al. [29] prepared a series of mixed crystal $TiO_2$ by changing the ratio of tartaric acid: $TiCl_3$. The photocatalyst composed of 77% anatase and 23% rutile (tartaric acid: $TiCl_3$ = 0.1) shows the highest activity.

Abundant researches focus on controlling phase composition of anatase/rutile mixed crystal by heat treatment temperature. In this work, the influence of doping elements on the anatase→rutile phase transformation was adopted to regulate the phase composition at a fixed temperature. Sn and Ni elements doping were employed to modify $TiO_2$ and adjust the content of anatase and rutile. The effect of co–doping on the structure and photocatalytic property of anatase/rutile mixed $TiO_2$ were studied.

## 2. Results and Discussion

### 2.1. Crystal Structure

Figure 1 displays the XRD patterns of samples. The diffraction peaks of pure $TiO_2$ at 25.4°, 37.1°, 37.9°, 38.7°, 48.2°, 54.0°, 55.2°, and 62.7° correspond to the (101), (103), (004), (112), (200), (105), (211), and (204) crystal planes of anatase structure. Besides, a faint diffraction peak around 27.5° can be ascribed to the (110) plane of rutile structure, which confirms that pure $TiO_2$ is anatase/rutile mixed crystal structure. Compared with pure $TiO_2$, the anatase peak intensity of Sn–$TiO_2$ decreases, while the peak intensity of rutile increases. The rutile content is 41.1%, which is higher than that of pure $TiO_2$ (2.1%), indicating that the transformation from anatase to rutile was advanced by Sn doping. The crystal structure $SnO_2$ is similar to rutile, which can act as the nucleation center of rutile and accelerate the formation and growth process of rutile nucleus, thus promotes the phase transformation [31]. No peak of rutile is detected in Ni–$TiO_2$ and all the peaks are ascribing to anatase, which indicates that Ni doping inhibits the phase transformation [32]. The effect of doping on phase transformation may be related to the oxide fusing point of doped elements [33]. If the fusing point of the oxide is higher than $TiO_2$, it shows inhibition effect, and when it is lower than $TiO_2$, it promotes the phase transformation. The fusing point of $SnO_2$ (1127 °C) is lower than $TiO_2$ (1640 °C), and the fusing point of NiO is (1990 °C) higher than $TiO_2$. On the other hand, the radium of $Sn^{4+}$ and $Ni^{2+}$ (0.069 nm) is close to $Ti^{4+}$ radium (0.0605 nm), which makes $Sn^{4+}$ and $Ni^{2+}$ ions able to enter into the lattice to replace $Ti^{4+}$ ions, bringing more crystal defects and promoting the phase transformation [34]. In summary, Sn doping and Ni doping exhibit the promotion and suppression of phase transition, respectively. The rutile content of Sn/Ni–$TiO_2$ sample is 5.3%, which is higher than pure $TiO_2$, indicating that the promotion effect of Sn doping on phase transformation is greater than the inhibition effect of Ni doping. The crystalline size and phase composition of the obtained photocatalysts are listed in Table 1.

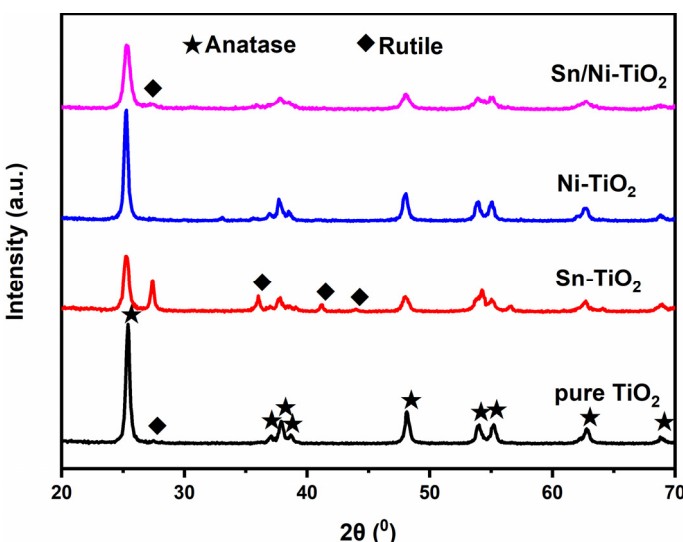

**Figure 1.** XRD patterns of samples.

**Table 1.** Phase composition and Crystallite size (D) of samples.

| Samples | Phase Composition (Anatase/Rutile) | D (nm) |
|---|---|---|
| pure TiO$_2$ | 97.9/2.1 | 21.5/26.9 |
| Sn–TiO$_2$ | 58.9/41.1 | 16.9/25.8 |
| Ni–TiO$_2$ | 100.0/0 | 21.7 |
| Sn/Ni–TiO$_2$ | 94.7/5.3 | 14.3/19.9 |

*2.2. Morphology*

Figure 2 shows the SEM images of samples. The pure TiO$_2$ is granular and the size of a single particle is 20 nm, approximately. The agglomeration is serious, and the sizes of the agglomerates range from 20–200 nm. Ni–TiO$_2$ and Sn/Ni–TiO$_2$ exhibit similar morphology to pure TiO$_2$. However, the particles in Sn–TiO$_2$ are looser, the agglomeration relieves, and the agglomerate size decreases.

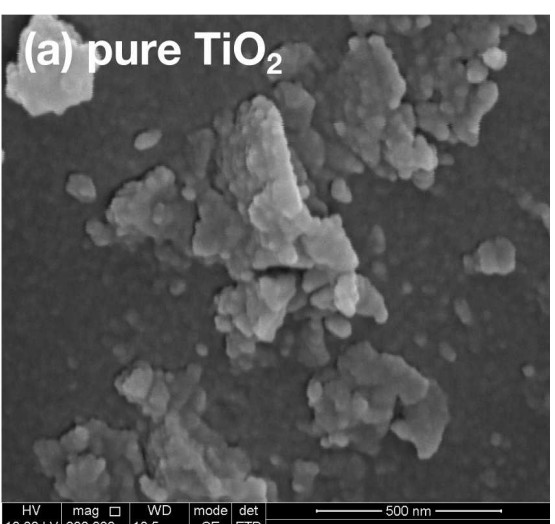
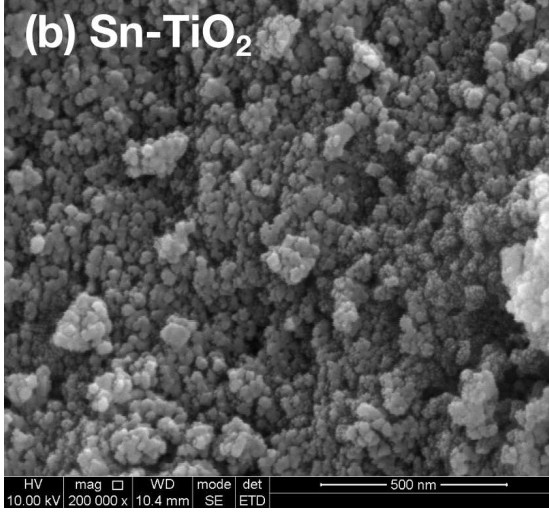

**Figure 2.** *Cont.*

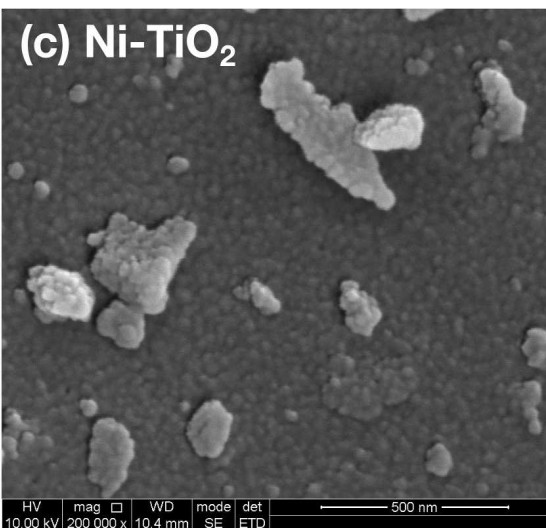

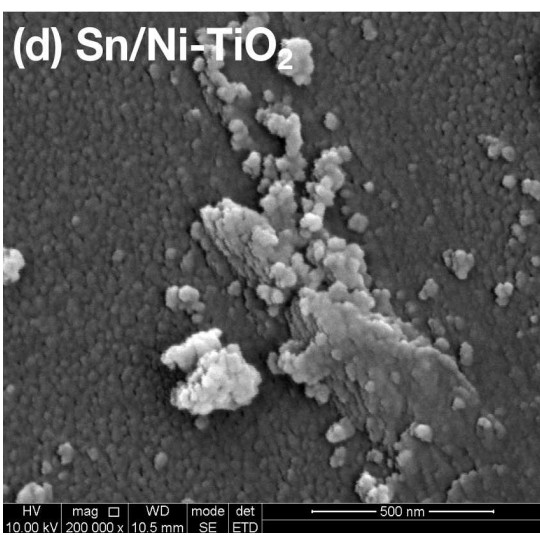

**Figure 2.** SEM images of samples: (**a**) pure $TiO_2$, (**b**) Sn–$TiO_2$, (**c**) Ni–$TiO_2$, (**d**) Sn/Ni–$TiO_2$.

The TEM and HRTEM images of pure $TiO_2$ (a, c) and Sn–$TiO_2$ (b, d) were exhibited in Figure 3. It is observed that the single particle size of pure $TiO_2$ is 20–30 nm, and the particle size of Sn–$TiO_2$ is smaller (15–20 nm). The crystal plane spacing marked in Figure 3c 0.348 nm can be attributed to the anatase (101) crystal plane. In Figure 3d, the marked crystal plane spacing is 0.355 nm, ascribing to the crystal plane of anatase (101), which is slightly increased compared to pure $TiO_2$ [20]. As the radius of $Sn^{4+}$ is larger than $Ti^{4+}$, $Sn^{4+}$ ions enter into $TiO_2$ lattice to replace $Ti^{4+}$ ions, causing lattice expansion and increasing the crystal plane spacing [35–37]. The interplanar spacing 0.325 nm in Figure 3d can be ascribed to the rutile (110) crystal plane, indicating that Sn–$TiO_2$ is mixed crystal structure [38]. This is consistent with XRD results.

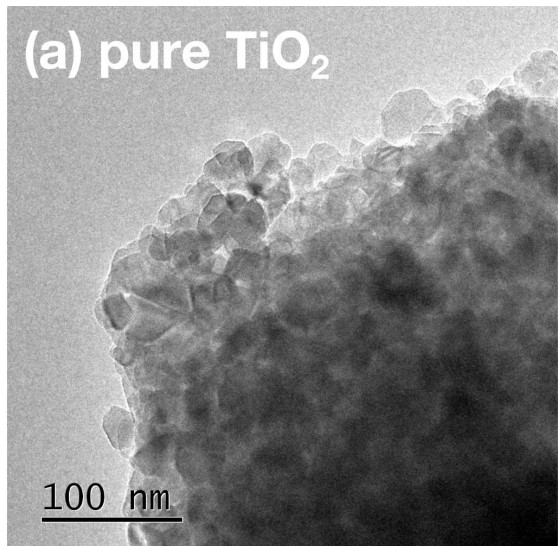

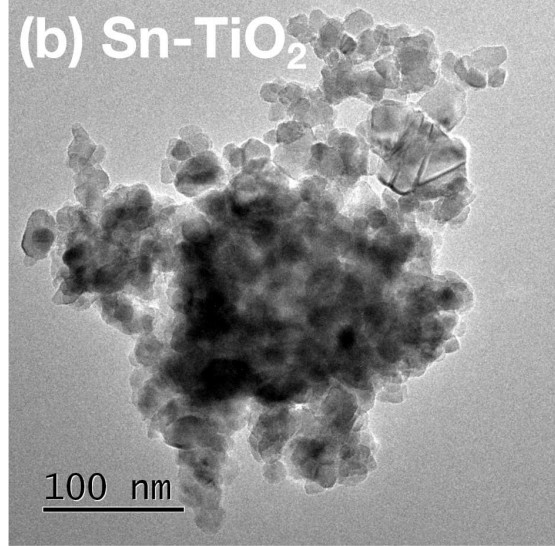

**Figure 3.** *Cont*.

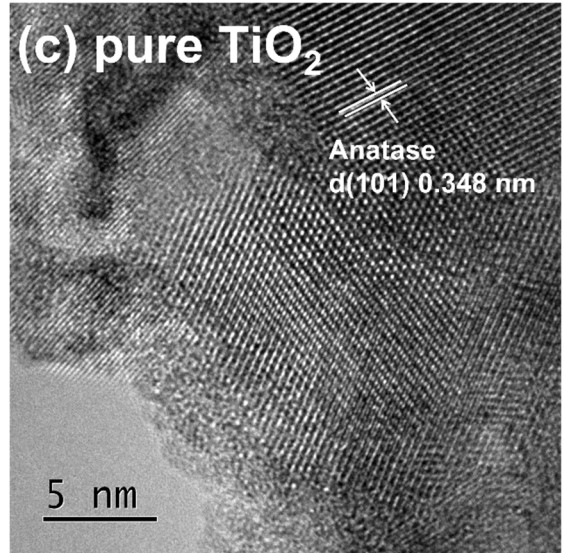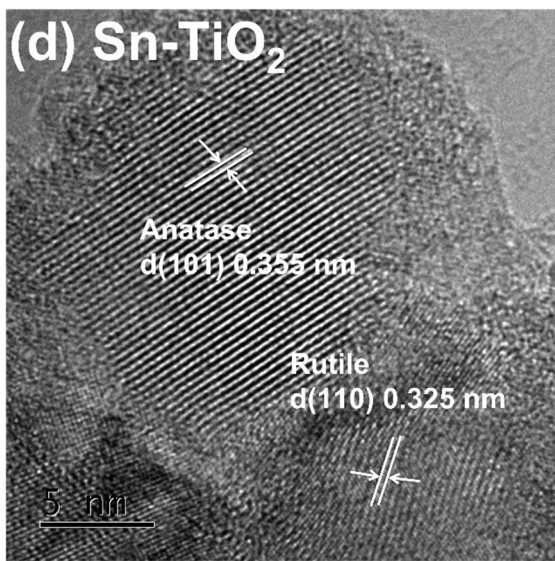

**Figure 3.** TEM and HRTEM images of pure TiO$_2$ (**a**,**c**) and Sn–TiO$_2$ (**b**,**d**).

### 2.3. Element Composition and State

Figure 4 presents the XPS spectra of Sn/Ni–TiO$_2$. The full spectrum displays that Sn/Ni–TiO$_2$ sample contains five elements: Ti, O, C, Sn, and Ni. Two characteristic peaks of Ti 2p located at 458.3 eV and 464.0 eV are attributed to Ti 2p$_{3/2}$ and Ti 2p$_{1/2}$. The distance between the two peaks is 5.7 eV, implying that Ti element exists as Ti$^{4+}$ [38,39]. The O 1s peak splits into two characteristic peaks at 529.6 eV and 531.0 eV, corresponding to lattice oxygen (O$^{2-}$) and surface hydroxyl (OH$^-$) [5,38]. The Sn 3d spectrum consists of two peaks at 486.1 eV and 494.6 eV, which are ascribed to Sn 3d$_{5/2}$ and Sn 3d$_{3/2}$, indicating that Sn element exists as Sn$^{4+}$ [35]. The peaks located at 855.7 eV and 861.8 eV correspond to Ni 2p$_{3/2}$ and the peaks located at 873.2 eV and 880.1 eV correspond to Ni 2p$_{1/2}$, suggesting that Ni element exists in the form of +2 valence [14,40,41].

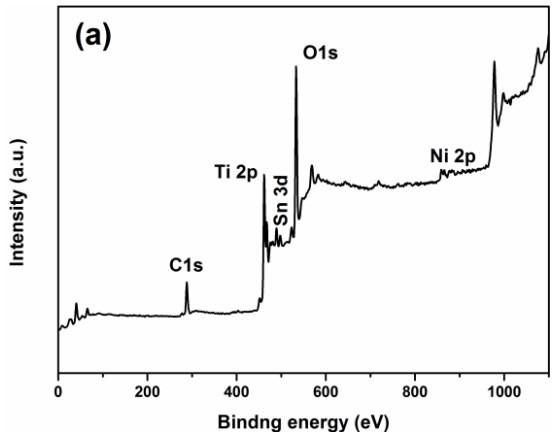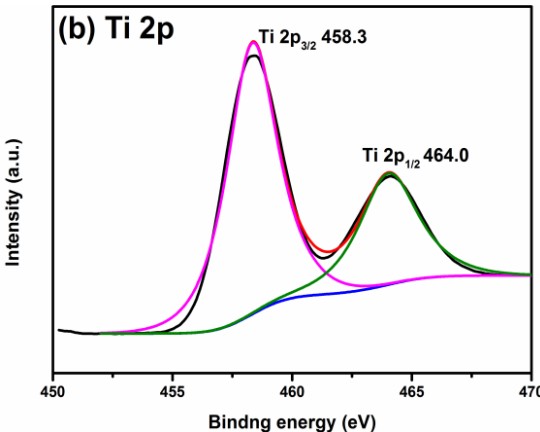

**Figure 4.** *Cont.*

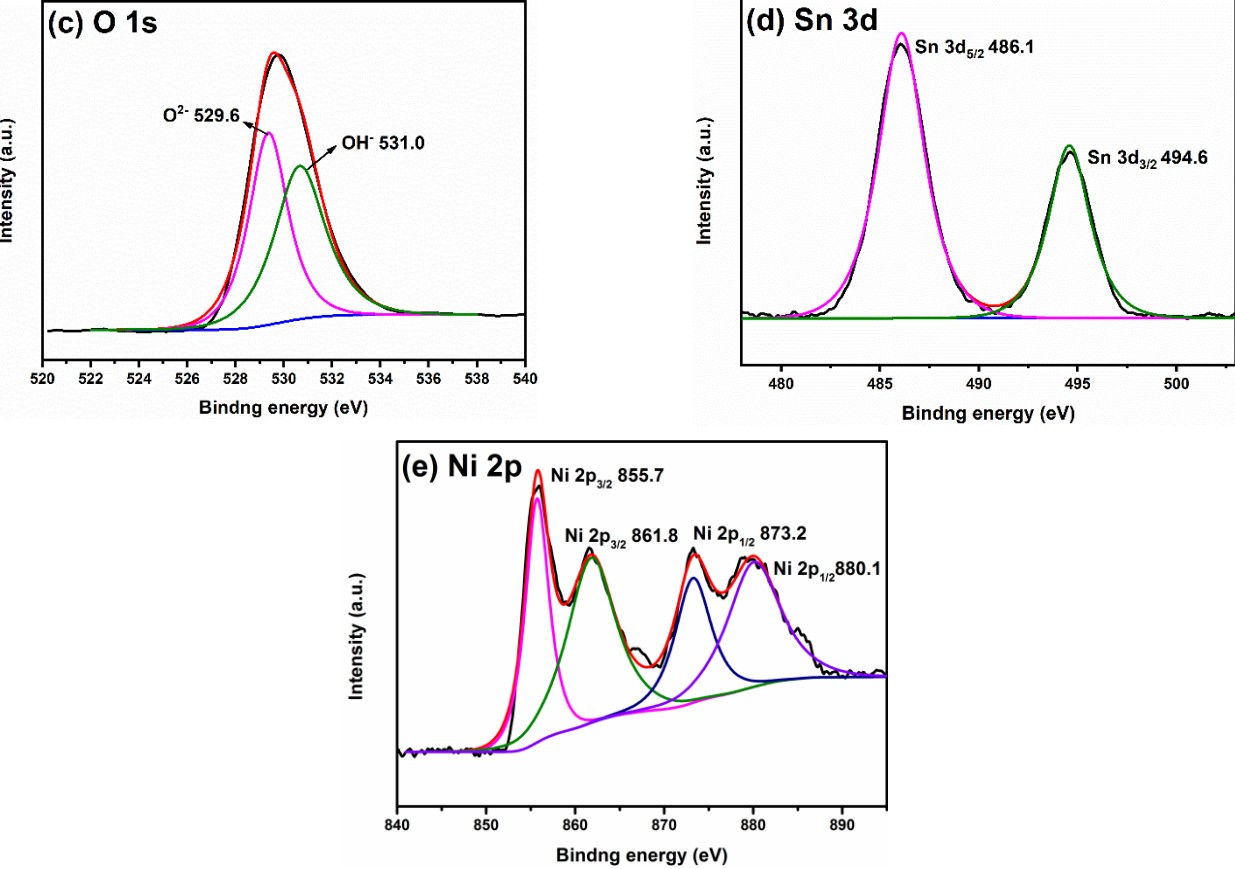

**Figure 4.** XPS spectra of Sn/Ni–TiO$_2$: (**a**) full spectrum, (**b**) Ti 2p, (**c**) O 1s, (**d**) Sn 3d, and (**e**) Ni 2p.

### 2.4. Optical Property

Figure 5 displays the PL spectra. The main peak of pure TiO$_2$ is around 400 nm, which is derived from the direct recombination of photogenerated electrons from conduction band back to valence band with holes. The PL peaks between 440–480 nm mainly originated from the recombination of photogenerated electrons in oxygen vacancies or crystal defects [42,43]. All of the doped samples show less PL peak intensity than pure TiO$_2$, indicating that both single doping and co-doping are beneficial to inhibiting the recombination of photogenerated charges. Crystal defects and oxygen vacancies are introduced by doping, which capture photoinduced charges, improving quantum efficiency. Remarkably, Sn–TiO$_2$ shows the lowest PL peak intensity. XRD and HRTEM results reveal that Sn–TiO$_2$ is anatase/rutile mixed phase structure, which is beneficial to the migration of photogenerated charges between phase interfaces [44]. Multi–doping produces a synergistic effect on introducing defects and inhibiting carrier recombination and improves quantum efficiency [15]. Nevertheless, the peak intensity of Sn/Ni–TiO$_2$ is less than Ni–TiO$_2$ but higher than Sn–TiO$_2$. The rutile ratio in Sn/Ni–TiO$_2$ is trace (5.3%), which makes the mixed crystal effect insufficient [21,26]. Therefore, Sn/Ni–TiO$_2$ is inferior to Sn–TiO$_2$ in inhibiting photogenerated charges recombination because the anatase/rutile phase composition of Sn–TiO$_2$ is suitable, which can reflect the mixed crystal effect to a large extent.

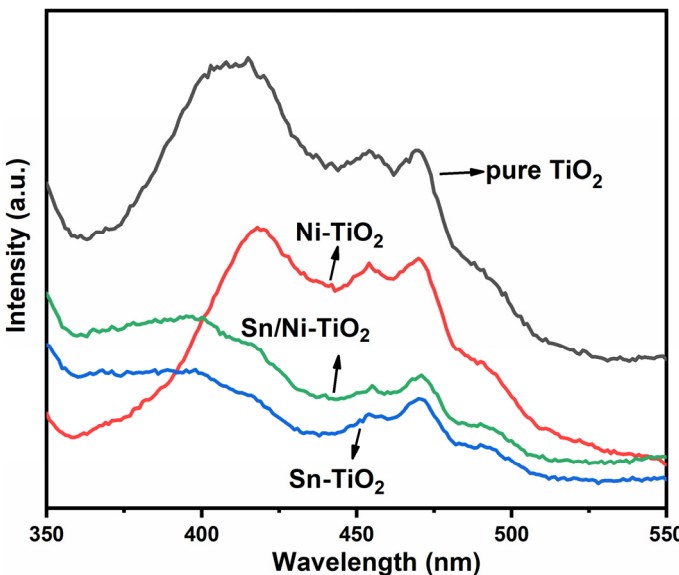

**Figure 5.** Photoluminescence (PL) spectra of pure TiO$_2$, Ni–TiO$_2$, Sn–TiO$_2$, and Sn/Ni–TiO$_2$.

Figure 6 shows the UV–Vis absorption spectra. The absorption of doped samples in the ultraviolet part is higher than that of pure TiO$_2$. Ni–TiO$_2$ shows a blue shift, while Sn–TiO$_2$ and Sn/Ni–TiO$_2$ show a red shift. Sn doping is in favor of the visible light utilization.

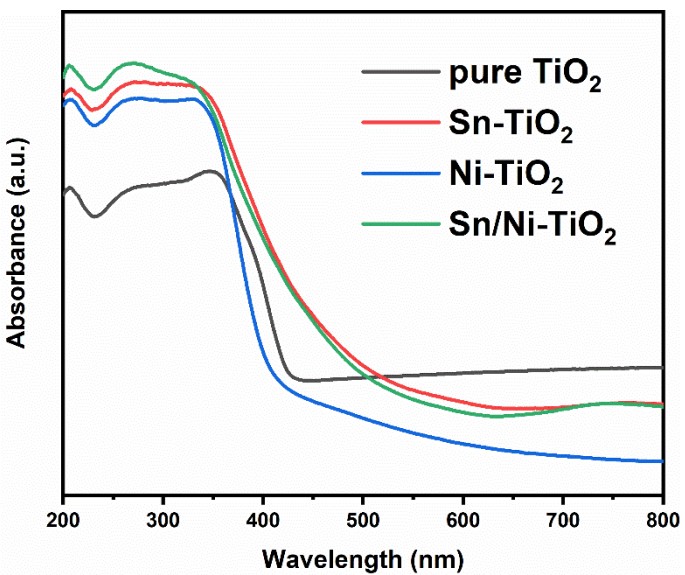

**Figure 6.** Ultraviolet–visible absorption spectra of pure TiO$_2$, Ni–TiO$_2$, Sn–TiO$_2$, and Sn/Ni–TiO$_2$.

### 2.5. Photocatalytic Activity

Figure 7a displays the decolorization degree curves of samples. Without catalyst, the decolorization degree of MB is 16.0% under irradiation after 150 min. The decolorization degree of pure TiO$_2$ is 23.6%, which is lower than Ni–TiO$_2$ (36.6%). The PL results prove that the recombination rate decreases after Ni doping, which is conducive to photocatalytic property. The decolorization degree of Sn–TiO$_2$ is 79.5%, which is significantly higher than pure TiO$_2$. Sn–TiO$_2$ exhibits the highest quantum efficiency and Sn doping improves the utilization of visible light, therefore, the photocatalytic property of Sn–TiO$_2$ is the best. The decolorization degree of Sn/Ni–TiO$_2$ is 48.6%, which is lower than Sn–TiO$_2$ but higher than Ni–TiO$_2$. This is in line with the PL results.

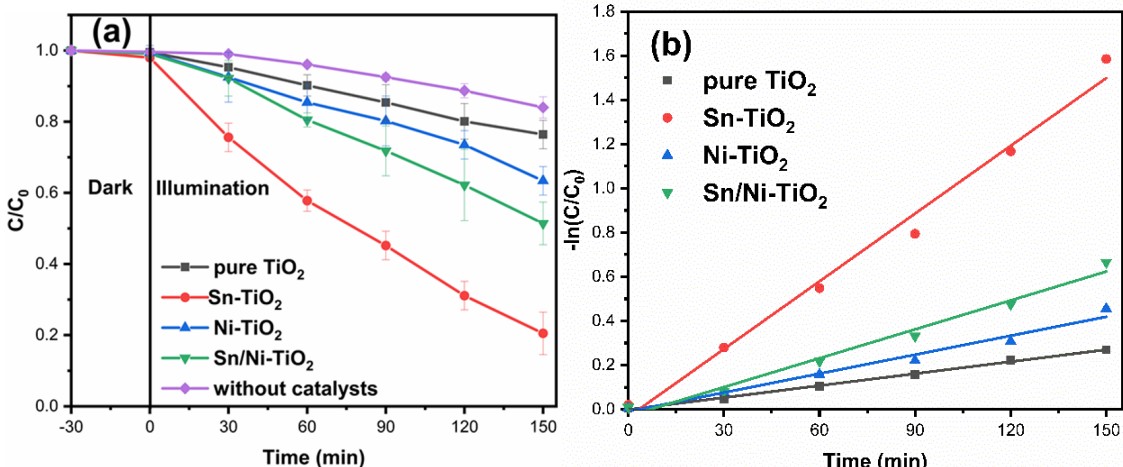

**Figure 7.** Decolorization degree curves (**a**) and kinetics curves (**b**) of samples.

Figure 7b shows the kinetics curve of samples. The decolorization of MB conforms to first–order reaction. The reaction rate constant k is computed using the formula $kt = -\ln (C_t/C_0)$ (where t represents the reaction time, $C_0$ represents the initial concentration of MB, and $C_t$ represents the concentration of MB at time t). The higher k is, the faster the reaction rate is. The calculated results show that the reaction rate constant of Sn–TiO$_2$ is 0.01022 min$^{-1}$, which is 5.6 times higher than pure TiO$_2$ (0.00181 min$^{-1}$).

*2.6. Mechanism of Photocatalytic*

Figure 8 is the schematic diagram of transfer path of the photogenerated charges in Sn–TiO$_2$. XRD and HRTEM results confirm that Sn–TiO$_2$ is an anatase/rutile mixed crystal structure. Electrons in valence band (VB) will be excited to conduction band (CB) to form photogenerated electrons when TiO$_2$ is exposed under light irradiation, leaving corresponding holes in VB. On the one hand, Sn doping introduces impurity energy level in forbidden band, reducing the excitation energy, promoting the utilization of light source. On the other hand, since the position of rutile CB is lower than anatase, the electrons in anatase CB will migrate to rutile CB, which speeds up the transfer of photoinduced electrons, prolongs the carrier life and improves the quantum efficiency [19–21]. The separated photogenerated electrons react with O$_2$ to generate superoxide free radicals ●O$_2$$^-$ and holes react with OH$^-$ to generate ●OH radicals. These free radicals and holes (h$^+$) decompose MB owing to their strong oxidation [19,45].

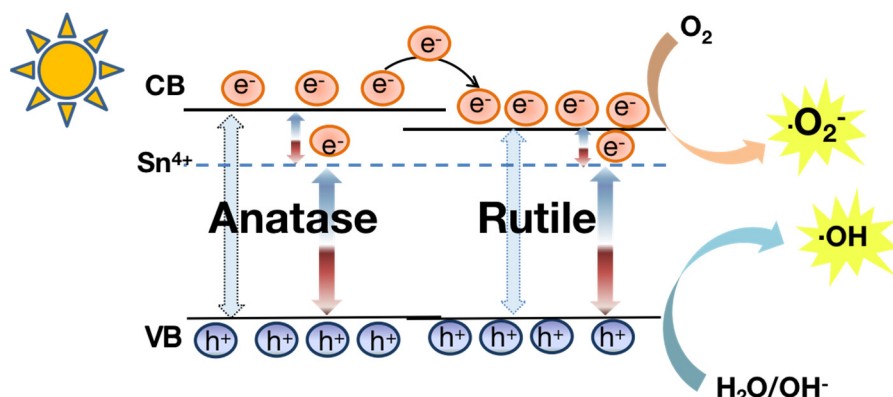

**Figure 8.** Schematic diagram of transfer path of the photogenerated charges in Sn–TiO$_2$.

## 3. Experimental

### 3.1. Sample Preparation

Anhydrous ethanol (Analytical Reagent, AR), butyl titanate (AR), glacial acetic acid (AR), stannic chloride (AR), nickel chloride (AR) and methylene blue (AR) were purchased from Chengdu Chron Chemicals Co., Ltd. (Chengdu, China).

Solution A was obtained by adding butyl titanate and absolute ethyl alcohol in a volume ratio of 7:10. Deionized water, anhydrous ethanol, and glacial acetic acid were added with a volume ratio of 1:6:2 to gain Solution B. The volume ratio of A:B is 17:9. Solution B was dropwise put into solution A to form a sol. After aging, the sol converted to gel, which was undergoing drying and calcining at 550 °C for 1 h to get pure $TiO_2$. A certain amount of $SnCl_4 \cdot 5H_2O$ or $NiCl_2 \cdot 6H_2O$ was put into solution B to prepare Sn-doped, Ni-doped, and Sn/Ni co-doped $TiO_2$. The molar ratios of Sn/Ti and Ni/Ti were both 3%. They were marked as $Sn–TiO_2$, $Ni–TiO_2$, and $Sn/Ni–TiO_2$.

### 3.2. Sample Characterization

The crystal structure of samples was analyzed by a DX–2700 X–ray diffractometer (Dandong Haoyuan Instrument Co. Ltd., Dandong, China). The test current was 30 mA, the voltage was 40 kV, and the scanning angle was 20°–70° with the scanning speed being 0.06°/s. The crystallite sizes (D) were computed by the Scherrer formula: $D = 0.89\lambda/\beta\cos\theta$, where $\lambda$ is the wavelength of Cu Ka, $2\theta$ is the Bragg diffraction angle, and $\beta$ is the full width at half maximum of the diffraction peak. The mass fraction of anatase ($X_A$) was computed by formula: $X_A = (1 + 1.26(I_R/I_A))^{-1}$, where $I_R$ and $I_A$ are the intensities of rutile (110) plane and anatase (101) plane. The morphology of samples (SEM and TEM) was observed using a Inspect F50 scanning electron microscope and a Tecnai G2 F20 transmission electron microscope (FEI Company, Hillsboro, OR, USA). The element composition and valence were analyzed by a multifunctional surface analysis system (XSAM800, Kratos Ltd., Manchester, Britain); The photoluminescence spectra were recorded on a fluorescence spectrometer (F–4600, Shimadzu Group Company, Kyoto, Japan); The optical absorption was tested using an ultraviolet-visible photometer (UV–3600, Shimadzu Group Company, Kyoto, Japan).

### 3.3. Photocatalysis Experiment

100 mL (10 mg/L) MB aqueous solution and 100 mg samples were mixed in a beaker. The obtained mixture was stirred in dark 30 min to achieve the adsorption and desorption equilibrium. Next, a 250 W xenon lamp with wavelength from 300 nm to 800 nm was turned on, which was placed 7.5 cm above the liquid level. The absorbance of the mixture was measured every 30 min after irradiation. The decolorization degree (D) was computed using the equation $D = (A_0 - A_t)/A_0 \times 100\%$.

## 4. Conclusions

Pure $TiO_2$, $Sn–TiO_2$, $Ni–TiO_2$, and $Sn/Ni–TiO_2$ nanomaterials were obtained through the sol–gel route. The phase transformation from anatase to rutile is advanced by Sn doping, while it is inhibited by Ni doping. $Sn–TiO_2$ is anatase/rutile mixed crystal structure, which accelerates the migration of photoelectric charges in phase interface, increases the lifetime of charge carriers, and improves the quantum efficiency. Besides, Sn doping is in favor of light absorption. $Sn/Ni–TiO_2$ is inferior to $Sn–TiO_2$ in photoinduced charges separation owing to its trace rutile ratio, which limits the mixed crystal effect. Therefore, the photocatalytic activity of $Sn–TiO_2$ is higher than $Sn/Ni–TiO_2$ and pure $TiO_2$. The first–order reaction rate constant of $Sn–TiO_2$ is 5.6 times higher than pure $TiO_2$.

**Author Contributions:** Methodology, Q.Q. and J.W.; investigation, Y.X., D.Y. and Q.Z.; supervision, X.Z.; project administration, W.F.; formal analysis, Q.Q. and X.Z.; funding acquisition, X.Z. All authors have read and agreed to the published version of the manuscript.

**Funding:** This study was funded by the Training Program for Innovation of Chengdu University (grant numbers: S202011079053, CDU-CX-2021527).

**Data Availability Statement:** Data is contained within the article.

**Conflicts of Interest:** The authors declare no conflict of interest.

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
