# Peer review of "Synthesis and Characterization of Sn/Ni Single Doped and Co–Doped Anatase/Rutile Mixed–Crystal Nanomaterials and Their Photocatalytic Performance under UV–Visible Light"

_catalysts, doi:10.3390/catal11111341_

Round 1

Reviewer 1 Report

Dear Authors,

your paper describes the sol-gel preparation and characterization of Ni- and Sn-doped titania. The effect on anatase-rutile conversion is investigated in correlation to the UV-decomposition of methylene blue.

For publication, the following remarks should be considered:

headline:

Syntheses and characterization….. 

Abstract:

Line 16, 78.6% makes no sense without the corresponding reaction time and concentration! …3.2 times…should be sufficient…or you have to explain more details!

Experimental:

2.1 Preparation:The ratio solutionA/solutionB is missed.

2.2 Sample Characterization

The machines used should be described more in detail, beside the name also the company should be noted. Also some specifications like x-ray tube, detector etc. And the sample preparation is missed, did you use the powder for all methods? How is the crystallite size and amount of anatase or rutile calculated? Software?

2.3 Photocatalysis:

Did you work with an aqueous MB solution? This should be noted!

What’s the wavelength of the xenon lamps? What’s the distance to the suspension? And which power reaches the suspension?

Results:

3.5 Photocatalytic activity

Fig. 7 and calculation of reaction rate constant k:

If I look to the slope in Fig.7b), for pure TiO2-samples I find k=0.009 and for Sn-TiO2-samples k=0.0017. So k is maybe 6 times higher, but not 9 times… A data table would be helpful to reconstruct the calculation!

3.1 Crystal structure

Line 95-102 Ref 31 is about Cu-doping, what will you say with this paper?

You think, that the melting point of the dopants is related to phase transition? Do you have any references to this assumption? Song et.al 2005 (Preparation and phase transformation of anatase–rutile crystals in metal doped TiO2/muscovite nanocomposites, thin solid films 491, 120-116) suggest, that the phase transition depends whether the dopant replaces Ti in the network or not. You should also discuss this point of view! In your XPS data you show, that Sn is 4+ and Ni is 2+!

And what’s about the particle size? The Sn-doped titania is smaller, so the surface area is higher and photocatalytic activity can be increased!

Author Response

List of Responses to the Comments-1447674

Reviewer 1

Your paper describes the sol-gel preparation and characterization of Ni- and Sn-doped titania. The effect on anatase-rutile conversion is investigated in correlation to the UV-decomposition of methylene blue.

For publication, the following remarks should be considered:

headline:

Syntheses and characterization…..

Abstract:

Line 16, 78.6% makes no sense without the corresponding reaction time and concentration! …3.2 times…should be sufficient…or you have to explain more details!

Experimental:

2.1 Preparation:The ratio solutionA/solutionB is missed.

2.2 Sample Characterization

The machines used should be described more in detail, beside the name also the company should be noted. Also some specifications like x-ray tube, detector etc. And the sample preparation is missed, did you use the powder for all methods? How is the crystallite size and amount of anatase or rutile calculated? Software?

2.3 Photocatalysis:

Did you work with an aqueous MB solution? This should be noted!

What’s the wavelength of the xenon lamps? What’s the distance to the suspension? And which power reaches the suspension?

Results:

3.5 Photocatalytic activity

Fig. 7 and calculation of reaction rate constant k:

If I look to the slope in Fig.7b), for pure TiO2-samples I find k=0.009 and for Sn-TiO2-samples k=0.0017. So k is maybe 6 times higher, but not 9 times… A data table would be helpful to reconstruct the calculation!

3.1 Crystal structure

Line 95-102 Ref 31 is about Cu-doping, what will you say with this paper?

You think, that the melting point of the dopants is related to phase transition? Do you have any references to this assumption? Song et.al 2005 (Preparation and phase transformation of anatase–rutile crystals in metal doped TiO2/muscovite nanocomposites, thin solid films 491, 120-116) suggest, that the phase transition depends whether the dopant replaces Ti in the network or not. You should also discuss this point of view! In your XPS data you show, that Sn is 4+ and Ni is 2+!

And what’s about the particle size? The Sn-doped titania is smaller, so the surface area is higher and photocatalytic activity can be increased!

  1. headline:

Syntheses and characterization…..

A: Thanks for the reviewer’s suggestion. According to the reviewer’s suggestion, the authors have revised the title as “Synthesis and characterization of Sn/Ni single doped and co-doped anatase/rutile mixed-crystal nanomaterials and their photocatalytic performance under UV-visible light”.

  1. Abstract:

Line 16, 78.6% makes no sense without the corresponding reaction time and concentration! …3.2 times…should be sufficient…or you have to explain more details!

A: Thanks for the reviewer’s high academic level comments. According to the reviewer’s suggestions, the authors have revised as “The degradation degree of methylene blue (MB) by Sn-TiO2 is 79.5% after 150 min. The reaction rate constant of Sn-TiO2 is 0.01022 min-1, which is 5.6 times higher than pure TiO2 (0.00181 min-1).”

  1. Experimental:

2.1 Preparation: The ratio solution A/solution B is missed.

A: Thanks for the reviewer’s suggestion. The volume ratio of A:B is 17:9.

2.2 Sample Characterization

The machines used should be described more in detail, beside the name also the company should be noted. Also some specifications like x-ray tube, detector etc. And the sample preparation is missed, did you use the powder for all methods? How is the crystallite size and amount of anatase or rutile calculated? Software?

A: Thanks for the reviewer’s high academic level comments. All the characterization methods were carried on the powder samples. According to the reviewer’s suggestions, the authors have added the company and detail of the machines. The crystal structure of samples was analyzed by a DX-2700 X-ray diffractometer (Dandong Haoyuan, China). The test current was 30 mA, the voltage was 40 kV, and the scanning angle was 20°-70° with the scanning speed was 0.06°/s. The crystallite sizes (D) were computed by the Scherrer formula: D = 0.89λ/βcosθ, where λ is the wavelength of Cu Ka, 2θ is the Bragg diffraction angle, and β is the full width at half maximum of the diffraction peak. The mass fraction of anatase (XA) was computed by formula: XA= (1 + 1.26(IR/IA))-1, where IR and IA are the intensities of rutile (110) plane and anatase (101) plane.

The morphology of samples (SEM and TEM) was observed using a Inspect F50 scanning electron microscope and a Tecnai G2 F20 transmission electron microscope (FEI, USA). The element composition and valence were analyzed by a multifunctional surface analysis system (XSAM800, Britain); The photoluminescence spectra were recorded on a fluorescence spectrometer (F-4600, Japan); The optical absorption was tested using an ultraviolet-visible photometer (UV-3600, Japan).

2.3 Photocatalysis:

Did you work with an aqueous MB solution? This should be noted!

A: Thanks for the reviewer’s high academic level comments. The target pollutant is MB aqueous solution.

What’s the wavelength of the xenon lamps? What’s the distance to the suspension? And which power reaches the suspension?

A: Thanks for the reviewer’s high academic level comments. According to the reviewer’s suggestion, the authors have confirmed the relevant parameters of the light source. The schematic illustration and the photograph of the photoreactor are shown in Fig. 1. The power intensity at the surface of the solution is 0.277 W/cm2. The wavelength of the light source is 300 nm-800 nm. The lamp was placed 7.5 cm above the liquid level.

Fig. 1 The schematic illustration and the photograph of the photoreactor.

  1. Results:

3.5 Photocatalytic activity

Fig. 7 and calculation of reaction rate constant k:

If I look to the slope in Fig.7b), for pure TiO2-samples I find k=0.009 and for Sn-TiO2-samples k=0.0017. So k is maybe 6 times higher, but not 9 times… A data table would be helpful to reconstruct the calculation!

A: Thanks for the reviewer’s high academic level comments. We repeatedly tested the degradation rate of MB and took its average to get Fig. 7. According to the reviewer’s suggestion, the authors have added a data table. The reaction rate constant of Sn-TiO2 is 0.01022 min-1, which is 5.6 times higher than pure TiO2 (0.00181 min-1).

Fig. 7 Degradation degree curves (a) and kinetic curves (b) of samples.

Table 1 Kinetic fitting data of samples

Samples

R2

K (min-1)

Pure TiO2

0.99681

0.00181

Sn-TiO2

0.98476

0.01022

Ni-TiO2

0.97171

0.00286

Sn/Ni-TiO2

0.97885

0.00436

3.1 Crystal structure

Line 95-102 Ref 31 is about Cu-doping, what will you say with this paper?

You think, that the melting point of the dopants is related to phase transition? Do you have any references to this assumption? Song et.al 2005 (Preparation and phase transformation of anatase–rutile crystals in metal doped TiO2/muscovite nanocomposites, thin solid films 491, 120-116) suggest, that the phase transition depends whether the dopant replaces Ti in the network or not. You should also discuss this point of view! In your XPS data you show, that Sn is 4+ and Ni is 2+!

And what’s about the particle size? The Sn-doped titania is smaller, so the surface area is higher and photocatalytic activity can be increased!

A: Thanks for the reviewer’s high academic level comments. In Ding's study, it was confirmed that the anataserutile phase transition is related to the melting point of doped element oxides, which is also a widely accepted view. This article has been cited more than 100 times. This article has been cited reference 33.

[33] X.Z. Ding, L. Liu, X. M. Ma, The influence of alumina dopant on the structural transformation of gel-derived nanometre titania powders[J]. Journal of Materials Science Letters, 13 (1994) 462-464.

As the reviewer point out, the ion radius has a great influence on whether the doped ions enter the lattice to replace Ti4+. The radium of Sn4+ and Ni2+ (0.069 nm) is close to Ti4+ radium (0.0605 nm), which makes Sn4+ and Ni2+ ions are able to enter into the lattice to replace Ti4+ ions, bringing more crystal defects and promoting the phase transformation. We will add the discussion in the revised manuscript. We have cited the article recommended by the reviewer as reference 34.

In addition, according to the XRD calculation results, the grain size of pure TiO2 anatase is reduced from 21.5 nm to 16.9 of Sn-TiO2. It can also be seen from SEM and TEM images that the particle size decreases after Sn doping. Smaller grain size and particle size are beneficial to improving the specific surface area and photocatalytic activity.

Thanks for the reviewer’s high academic suggestions and comments. Your suggestions and comments not only help us in improving the quality of our manuscript significantly, but also point out the scientific and innovative direction for our future research.

Reviewer 2 Report

Feng et al., in their paper, described the synthesis and characterization of Sn/Ni-doped anatase/rutile materials and their photocatalytic performance.

I am not particularly skilled in materials synthesis and characterization, so I would only comment on the photocatalysis section.

1/ The authors compare pure TiO2 (figure 7) synthesized by a hydrothermal process. The comparison with the reference material (Degussa P25) synthesized by flame hydrolysis of TiCl4 could be interesting. This comparison could justify the use of this new photocatalyst instead of the commercial product. A comparison with a blank experiment (no photoactive semiconductor see point 3/) should also be provided.

2/ Could the authors give the errors bars associated with the ratio of absorbance in figure 7. What is wavelength studied? What is the absorbance of a solution of MB at 10mg/L? All this critical data should be given, and a comparison of the UV spectra in the experimental section will be welcome.

3/ The term “degradation” line 167 and “photodegradation” line 176, as well as the CO2 and water as products of the so-called photodegradation in figure 8, are overstated.

The discoloration of MB solution does not imply that MB has been degraded. The discoloration of methylene blue in an aqueous solution has been observed without the presence of any photoactive semiconductor, simply by visible light irradiation (photosensitization process). The formed species are partially defragmented MB, i.e., leuco dyes, demethylated phenothiazine dyes, ...

Samples should be analyzed with LC-MS and ion chromatography to demonstrate the total degradation of MB (C16H18ClN3S) into CO2, nitrate and sulfate, and chloride.

In conclusion, the paper is stimulating and could be accepted after major revision.

Author Response

List of Responses to the Comments-1447674

Reviewer 2

Feng et al., in their paper, described the synthesis and characterization of Sn/Ni-doped anatase/rutile materials and their photocatalytic performance. I am not particularly skilled in materials synthesis and characterization, so I would only comment on the photocatalysis section.

  1. The authors compare pure TiO2 (figure 7) synthesized by a hydrothermal process. The comparison with the reference material (Degussa P25) synthesized by flame hydrolysis of TiCl4 could be interesting. This comparison could justify the use of this new photocatalyst instead of the commercial product. A comparison with a blank experiment (no photoactive semiconductor see point 3/) should also be provided.
  2. Could the authors give the errors bars associated with the ratio of absorbance in figure 7. What is wavelength studied? What is the absorbance of a solution of MB at 10mg/L? All this critical data should be given, and a comparison of the UV spectra in the experimental section will be welcome.
  3. The term “degradation” line 167 and “photodegradation” line 176, as well as the CO2 and water as products of the so-called photodegradation in figure 8, are overstated.

The discoloration of MB solution does not imply that MB has been degraded. The discoloration of methylene blue in an aqueous solution has been observed without the presence of any photoactive semiconductor, simply by visible light irradiation (photosensitization process). The formed species are partially defragmented MB, i.e., leuco dyes, demethylated phenothiazine dyes, ...

Samples should be analyzed with LC-MS and ion chromatography to demonstrate the total degradation of MB (C16H18ClN3S) into CO2, nitrate and sulfate, and chloride.

In conclusion, the paper is stimulating and could be accepted after major revision.

  1. The authors compare pure TiO2 (figure 7) synthesized by a hydrothermal process. The comparison with the reference material (Degussa P25) synthesized by flame hydrolysis of TiCl4 could be interesting. This comparison could justify the use of this new photocatalyst instead of the commercial product. A comparison with a blank experiment (no photoactive semiconductor see point 3/) should also be provided.

A: Thanks for the reviewer’s high academic level comments. According to the reviewer’s suggestion, the author conducted a P25 experiment, and the results showed that the photocatalytic activity of P25 was higher than that of the photocatalyst prepared in this experiment.

In addition, the author also conducted a blank experiment. Without the addition of photocatalyst, the decolorization degree of MB was 16.0% after 150 min.

  1. Could the authors give the errors bars associated with the ratio of absorbance in figure 7. What is wavelength studied? What is the absorbance of a solution of MB at 10mg/L? All this critical data should be given, and a comparison of the UV spectra in the experimental section will be welcome.

A: Thanks for the reviewer’s high academic level comments. According to the reviewer’s suggestion, the author tested the decolorization degree of MB for many times, took the average value of absorbance as the final data. The results are shown in Fig. 1.

Fig. 1 Degradation degree curves (a) and kinetic curves (b) of samples.

The schematic illustration and the photograph of the photoreactor are shown in Fig. 2. The wavelength of the light source is 300 nm-800 nm. The lamp was placed 7.5 cm above the liquid level. The absorbance of MB solution (10 mg/L) is 0.712 (664 nm).

Fig. 2 The schematic illustration and the photograph of the photoreactor.

  1. The term “degradation” line 167 and “photodegradation” line 176, as well as the CO2 and water as products of the so-called photodegradation in figure 8, are overstated.

The discoloration of MB solution does not imply that MB has been degraded. The discoloration of methylene blue in an aqueous solution has been observed without the presence of any photoactive semiconductor, simply by visible light irradiation (photosensitization process). The formed species are partially defragmented MB, i.e., leuco dyes, demethylated phenothiazine dyes, ...

Samples should be analyzed with LC-MS and ion chromatography to demonstrate the total degradation of MB (C16H18ClN3S) into CO2, nitrate and sulfate, and chloride.

A: Thanks for the reviewer’s high academic level comments. According to the reviewer's suggestion, we carried out a blank experiment to study the decolorization degree of MB without photocatalyst. Without catalyst, the degradation degree of MB was 16.0% under irradiation after 150 min. For Sn-TiO2, the degradation rate is 79.5%, so most of the decolorization of MB comes from the reaction of photocatalyst. The author replaced degradation with decolorization in the revised manuscript.

Thanks for the reviewer’s high academic suggestions and comments. Your suggestions and comments not only help us in improving the quality of our manuscript significantly, but also point out the scientific and innovative direction for our future research.

Round 2

Reviewer 2 Report

In their response, the authors state that an experiment was performed with the reference photocatalyst Degussa P25 and the results showed that the photocatalytic activity of P25 was higher than that of the photocatalyst prepared in this experiment. However, I could not find the data in figure 7 in the revised version, whereas a blank experiment has been provided as requested.

The experiment with P25 must be added in figure 7 and discussed in the text.

I also noticed in their answer that the authors took the average value of absorbance as the final data, but the error bars are still missing from figure 7.

I appreciate that the authors replaced the term “degradation” with “decolorization” in the revised manuscript. However, figure 8 must be modified accordingly. The authors must demonstrate that MB is oxidized into CO2, which has not been proven.

Author Response

  1. The experiment with P25 must be added in figure 7 and discussed in the text.

A: Thanks for the reviewer’s high academic level comments. According to the reviewer’s suggestion, the author conducted a P25 experiment, and the results shows the decolorization degree of P25 towards P25 is 100% after 60 min, which shows higher than that of the photocatalyst prepared in this experiment.

Fig. 7 Decolorization degree curves of samples and P25.

Actually, many studies did not list the comparison between the photocatalysts prepared in their study and P25. Meanwhile, some studies have compared with P25, and in these studies, the activity of the photocatalysts they prepared is higher than P25, which is conducive to enhance the value of their research. In our previous research, photocatalysts with higher activity than P25 have also been prepared. (RSC Adv., 2021, 11, 27257)

Here the authors would like to explain that the focus of this article is to study the effects of single doping and co-doping of Sn and Ni on the structure and photocatalytic properties of the anatase/rutile mixed crystal TiO2. The author believes that since the decolorization rate of the prepared photocatalyst is much lower than that of P25, it is not appropriate to include the data of P25 in the manuscript. The authors appreciate your understanding.

  1. I also noticed in their answer that the authors took the average value of absorbance as the final data, but the error bars are still missing from figure 7.

A: Thanks for the reviewer’s high academic level comments. According to the reviewer’s suggestion, the author have added the error bars in Fig. 7.

Fig. 7 Decolorization degree curves of samples

  1. I appreciate that the authors replaced the term “degradation” with “decolorization” in the revised manuscript. However, figure 8 must be modified accordingly. The authors must demonstrate that MB is oxidized into CO2, which has not been proven.

A: Thanks for the reviewer’s high academic level comments. The photocatalytic performance of photocatalysts can be evaluated by the change in target contaminant concentration. According to the Lambert-Beer law, the absorbance of the MB solution is proportional to the concentration of the solution in the lower concentration range, so the change in the measured absorbance can reflect the change in concentration. It is sufficient to assess the photocatalytic performance of photocatalysts by measuring the absorbance change degradation of target pollutants.

The measurement LC-MS and ion chromatography recommended by Review are suitable for studying decomposition products. Regrettably, the authors do not have the conditions to carry out this work. Therefore, we have modified Fig. 8 accordingly, focusing on the transfer path of the photogenerated charges.

The authors deeply appreciate your advice and understanding.
